# Preparation, Radiolabeling with ^68^Ga/^177^Lu and Preclinical Evaluation of Novel Angiotensin Peptide Analog: A New Class of Peptides for Breast Cancer Targeting

**DOI:** 10.3390/ph16111550

**Published:** 2023-11-02

**Authors:** Subhani M. Okarvi

**Affiliations:** Cyclotron and Radiopharmaceuticals Department, King Faisal Specialist Hospital and Research Centre, MBC-03, P.O. Box 3354, Riyadh 11211, Saudi Arabia; sokarvi@kfshrc.edu.sa; Tel.: +966-11-442-4812

**Keywords:** angiotensin, breast cancer, radiolabeling, peptide synthesis, biodistribution, preclinical

## Abstract

Aim: Angiotensin II (AngII) is known to play a significant part in the development of breast cancer by triggering cell propagation of breast cancer, tumor angiogenesis, and regulating tumor invasion and cell migration. AngII arbitrates its action via two G-protein-coupled receptors, AngII type 1 receptor (AT1) and AngII type 2 receptor (AT2). Overexpression of the AT1 receptor in breast cancer cells seems to promote tumor growth and angiogenesis, thus targeting the AT1 receptor using AngII peptide would facilitate the detection of breast carcinoma. We developed an AngII peptide intending to assess whether the peptide of the renin–angiotensin system holds the ability to target AT1 receptor-overexpressing breast cancer in vivo. Methods: DOTA-coupled AngII peptide was synthesized by conventional solid-phase peptide synthesis according to Fmoc/HATU chemistry. ^68^Ga/^177^Lu labeled AngII peptide was evaluated for its binding with TNBC MDA-MB-231 and ER+ MCF7 cell lines. Pharmacokinetics was studied in healthy balb/c mice and in vivo tumor targeting in nude mice with MDA-MB-231 tumors xenografts. Results: DOTA-AngII peptide was labeled efficiently with ^68^Ga/^177^Lu with high labeling efficiency (≥90%). The stability of the radiopeptide in human plasma was found to be high. The AngII peptide analog showed nanomolar (<40 nM) AT1 receptor-specific binding affinity. The radioactivity internalized into MDA-MBA-231 and MCF7 cells were 14.97% and 11.75%, respectively. In vivo, biodistribution in balb/c mice exhibited efficient clearance of ^68^Ga/^177^Lu-DOTA-AngII peptide from the blood and elimination predominantly by the renal system due to its hydrophilic nature. A low amount of radioactivity was seen in the major organs including lungs, liver, stomach, spleen, and intestines (<3% ID/g) except the kidneys. A high renal-urinary excretion was observed for the radiotracer. In the TNBC MDA-MB-231 xenografts model, radiolabeled AngII peptide exhibited specific and effective AT1-based targeting in vivo. A rapid and efficient tumor targeting (2.18% ID/g at 45 min p.i.) together with fast renal excretion (~67% ID) highlights the tumor-targeting potential of the radiotracer. The AT1 receptor specificity of the radiotracer was validated by blocking assays. Furthermore, PET imaging provided sufficient visualization of MDA-MB-231 tumors in nude mice. Conclusion: Our findings suggest that ^68^Ga/^177^Lu-DOTA-AngII peptide can be useful for the theranostic application of breast carcinomas. This study suggests the potential of this innovative class of peptides for rapid and efficient targeting of tumors and warrants further evaluation.

## 1. Introduction

In today’s world, breast cancer continues to be the most frequent cause of cancer death in women, thus necessitating further characterization of the mechanism of cancer advancement but also improving the efficiency of prognostic markers. It has been demonstrated that epithelial cells of the human breast show elements of the renin–angiotensin system (RAS) and studies advocate that these may be changed in disease development. Furthermore, any change in integrin expression is associated with lymph node metastasis [1,2,3,4].

Modern molecular imaging techniques, especially the positron-emission tomography (PET)-based nuclear imaging approach, provide the power of accurate image quantification with high sensitivity resulting in the precise diagnosis of the disease at the cellular level and ultimately resulting in better and improved treatment options, particularly in nuclear oncology. Similar importance is the finding of new and efficient molecular probes, such as small peptides holding the ability to bind tightly and specifically to the peptide receptors overexpressed on cancer cells allowing correct diagnosis of the diseases [5].

Angiotensin peptides, such as angiotensin (1–7) and angiotensin II (AngII), are endogenous 7–amino acid main effector peptides of the RAS, produced mainly from AngII by the enzymatic activity of angiotensin-converting enzyme 2 (ACE 2). It has been involved in several important features of cancer development, such as stimulating cell proliferation, migration, invasion, angiogenesis, and metastasis. AngII mediates its action employing two different G protein-coupled receptors superfamily, AngII type 1 receptor (AT1) and AngII type 2 receptor (AT2), which possess diverse biological functions. AT1 receptor arbitrates the majority of the actions of AngII, thus suggesting a useful target for the treatment of numerous diseases [6,7]. Moreover, it has been recently revealed that the high expression of the AT1 receptor in breast cancer cells causes epithelial–mesenchymal transition (EMT) and facilitates tumor growth and angiogenesis. EMT is known to be involved in advancing cancer cell invasion, cell spread, and chemoresistance [8,9]. Since angiotensin receptors are generally dispersed in epithelia, their probable importance to cancer is obvious. The recent findings further support that AngII is categorized as a RAS peptide possessing anti-proliferative properties, which hold the ability to inhibit many tumors, especially breast carcinomas. Additionally, studies have shown that AngII reduces the growth of TNBC cell lines in vitro as well as in vivo breast cancer cells growing in mice. Angiotensin treatment of mice models carrying TNBC resulted in a significant reduction in tumor size with no major side effects [1,2,3,4,6,10]. In addition, ACE inhibitors or angiotensin receptors displayed the ability to decrease tumor growth, angiogenesis, and metastasis in experimental mouse models in vivo [6,7]. The significant involvement of AngII in breast cancer growth makes it a clinically relevant target for breast cancer targeting.

It has been demonstrated that the presence of angiotensin hormone in the human body is usually related to blood pressure management. Recently, it has been revealed that angiotensin hormone also shrinks the progression of triple-negative breast cancer (TNBC) cells in vitro as well as in vivo tumors growing in mice, with no obvious harmful effects and the treatment resulted in a significant reduction in tumor mass [10,11,12]. Several studies propose that angiotensin peptides perhaps could be a different treatment strategy for breast cancer patients, by suppressing the tumor cell growth itself or by reducing the number of blood vessels or matrix proteins that deliver nutrients necessary for tumor cell growth [10,11,12]. Previous studies of ^125^I-angiotensin II binding using human breast cancer cells provided a high binding affinity, with a *K*_d_ value of ~60 nM [13]. Taken together, it seems logical to prepare an AngII peptide analog and explore its ability as a breast cancer targeting agent. It is anticipated that the noninvasive imaging of the expression of the AT1 receptor using radiolabeled AngII peptide analog would facilitate the timely diagnosis of breast cancer and possible monitoring of cancer treatment efficacy.

Amongst the most commonly used macrocyclic bifunctional ligands for preparing novel metal-based theranostic (imaging and therapeutic) agents is DOTA (1,4,7,10-tetraazacyclododecane-1,4,7,10-tetraacetic acid). DOTA chelator holds the distinct ability to form thermodynamically stable and kinetically inert complexes with both diagnostic PET isotope, gallium-68 (^68^Ga), and a reactor-produced low-energy β-emitter lutetium-177 (^177^Lu), thus providing “diagnose and therapy” theranostic strategy for the effective management the malignant tumors [5].

In our long interest in developing a peptide-based agent for targeting breast cancer, in this work, we aimed to explore the AngII peptide analog as a possible candidate for imaging of breast cancer. We have produced a novel AngII-based peptide by solid-phase synthesis and attached it to a DOTA chelating agent to promote radiolabeling with both ^68^Ga and ^177^Lu for possible theranostic use. Here, we describe the preparation, radiolabeling, and preclinical evaluation of a novel AngII-based tumor-targeting peptide.

## 2. Results and Discussion

The goal of this study was to investigate the effectiveness of the AngII peptide analog radiolabeled with ^68^Ga- and ^177^Lu using DOTA as a chelating group for targeting AT1 receptors overexpressed on breast carcinoma. The present paper is the first to describe the synthesis and preclinical evaluation of ^68^Ga/^177^Lu-labeled AngII peptide analog. Here we report the peptide synthesis, radiolabeling with both ^68^Ga and ^177^Lu, and in vitro and in vivo evaluations to examine its ability to target MDA-MB-231 and MCF-7 breast cancer cell lines in preclinical settings. It is anticipated that the formulation of AngII-based peptide analogs can be beneficial for the efficient targeting of human breast carcinomas.

### 2.1. Synthesis of the AngII Peptide

The AngII peptide analog was synthesized on a Rink amide resin according to the standard Fmoc peptide synthesis protocols. The desired amino acids were attached to the resin stepwise using the HATU/DIEA coupling methodology. Metal chelating agent, DOTA was attached as an amino acid and then the final peptide construct was removed from the resin and purified with the help of RP-HPLC. AngII peptide sequence: DOTA-Lys^13^-β-Ala^12^-Asp^11^-Arg^10^-Val^9^-Tyr^8^-Ile^7^-His^6^-Pro^5^-Phe^4^-Trp^3^-Lys^2^-Tyr^1^-CONH_2_ was extended at the C-terminus to include 3 more amino acids as a linker, Trp, Lys, and Tyr. These 3-amino acids were selected as the part of urotensin pharmacophore known to be important for breast cancer targeting [14]. Starting with 0.1 mmol of resin (220 mg), we obtained 70 mg of the DOTA-coupled AngII product in 33% overall yield based on the resin substitution rate. The peptide was purified by HPLC and its structural identity was confirmed by mass spectrometry analysis. Calculated monoisotopic mass: 2108; found: [M + 2H]^2+^ = 1055.

### 2.2. Radiolabeling of AngII Peptide with ^68^Ga and ^177^Lu

DOTA-coupled AngII peptide analog was radiolabeled efficiently with ^68^Ga just by mixing ~25 μg of the peptide with 2.5 M sodium acetate buffer and [^68^Ga]GaCl_3_ followed by heating to enhance the radiolabeling kinetics. A nearly similar method was used for radiolabeling of AngII peptide analog with ^177^Lu but in this case, 40 µL gentisic acid (40 mg/mL aqueous solution) was added to the reaction mixture to minimize the possible radiolysis. These facile procedures provide reproducible and efficient radiolabeling (greater than 85%) of the AngII peptide via DOTA chelator with both ^68^Ga and ^177^Lu with a molar radioactivity of higher than 300 Ci/mmol. It is evident from the radio-HPLC analyses that ^68^Ga/^177^Lu-labeled AngII peptide analog formed mainly one radioactive product (Figure 1), with retention times of 17.38 min and 18.06 min, respectively. Minor peaks of free ^68^Ga and free ^177^Lu were eluted from the column at 3.2 min and 3.4 min, respectively. The ^68^Ga/^177^Lu-complexes of AngII peptide analog were found to be stable for at least 4 h and 24 h, respectively as revealed by radio-HPLC analysis. Furthermore, the identification of ^68^Ga-DOTA-AngII peptide complex was achieved by comparison of the retention time with the cold ^nat^Ga-DOTA-AngII compound eluted with the same retention time under the same HPLC condition. Representative HPLC elution profiles are presented in Figure 1.

### 2.3. In Vitro Metabolic Stability in Human Plasma

High proteolytic stability is an important aspect of the tumor-targeting peptides to deliver the highest possible tumor-targeting effect. The extent of proteolytic degradation of the ^177^Lu-AngII peptide was determined by incubating the radiolabeled peptide with human plasma at 37° C for up to 5 h. The radio-HPLC evaluation showed that greater than 90% of the radioactivity was still associated with the radiolabeled peptide after 5 h of incubation. A limited release of free ^177^Lu amount (up to 10%) indicates the high in vitro plasma stability of the radiolabeled peptide, with low enzymatic breakdown by plasma proteases (Figure 2).

### 2.4. In Vitro Tumor Cell Binding

The ^68^Ga/^177^Lu-labeled AngII peptide analog was investigated for its capability to bind with AT1 receptor-expressing breast cancer cells. Triple-negative MDA-MB-231 and estrogen receptor-positive MCF7 breast cancer cell lines were used to determine saturation binding and the binding affinity (*K*_d_) profile was calculated by fitting the data with nonlinear regression using Graph-Pad Prism version 5.03 (GraphPad Software Inc., San Diego, CA, USA). The results are outlined in Table 1.

The cell binding data revealed that the ^68^Ga-labeled AngII peptide exhibited nanomolar level binding affinity (less than 40 nM) to the two breast cancer cell lines investigated in this work. The binding affinity (*K*_d_) values for MDA-MB-231 and MCF7 were found to be 30.0 ± 6.88 nM and 39.25 ± 8.10 nM, respectively (Table 1). The binding affinity pattern was found to be the same for the ^177^Lu-labeled AngII peptide but the values were somewhat better than the ^68^Ga-labeled AngII peptide. The values were 27.73 ± 5.97 nM for MDA-MB-231 and 35.14 ± 7.09 nM for the MCF7 breast cancer cell line. These bindings were significantly blocked when 200-fold molar excess unlabeled AngII peptide was added suggesting that the binding is receptor-specific. An overall tendency of somewhat higher binding affinity was seen (range, 27 nM to 30 nM) for both the ^68^Ga/^177^Lu-labeled AngII peptide to the triple-negative MDA-MB-231 over the estrogen receptor-positive MCF7 breast cancer cell line. In general, the binding of ^68^Ga/^177^Lu-labeled AngII peptide analog to breast cancer cells was saturable and receptor-specific. It is noteworthy here that the binding affinities of ^68^Ga/^177^Lu-labeled AngII peptide analog to the human breast cancer cell lines were found to be about two-fold higher than the previously reported for the ^125^I-angiotensin II binding to human breast carcinoma cells, holding the binding affinity (*K*_d_) value of ~60 nM [13].

For cellular internalization, the cell-bound radioactivity fraction was further incubated with an acidic buffer for 15 min at 37 °C to differentiate the cell-surface bound from the internalized radiolabeled peptide [15]. The data suggest that a fast and substantial degree of internalization for the ^177^Lu-labeled AngII peptide analog was observed with, 14.97 ± 5.03%, and 11.75 ± 3.70% of the cell-surface bound radiolabeled peptide internalized into MDA-MB-231 and MCF7 cells, respectively (Table 1). The findings of cell-binding and internalization into breast cancer cells show that the AngII peptide analog retained its maximum effectiveness and holds a high level of tumor cell affinity and specificity to AT1 receptor-expressing breast cancers. This highlights the usefulness of AngII peptide analog for targeting breast carcinomas.

### 2.5. Biodistribution Studies

The in vivo biodistribution of ^68^Ga-labeled AngII peptide analog was carried out in female Balb/c mice, at various time points, to observe the uptake in the normal tissues and organs and find the excretion route of the radiolabeled peptides. The biodistribution data are summarized in Table 2. For the ^68^Ga-labeled AngII peptide analog, a fast clearance from the blood circulation was observed both at 45 min and 2 h p.i., with 0.61% ID/g of radioactivity retained in the blood after 2 h p.i. Low uptake and retention of the radiopeptide in the liver varying from 1.54 ± 0.28% ID/g at 45 min to 0.48 ± 0.17% ID/g at 2 h p.i. is probably related to the high hydrophilicity (log *p* = −0.55 ± 0.11) of the ^68^Ga-labeled AngII peptide analog. The radioactivity found in the intestines, devoid of the contents, was also low (below 1.5% ID/g) both at 45 min and 2 h p.i. Nonetheless, a higher accumulation of radioactivity (up to 2.97% ID/g) was noticed when the whole intestines were measured. A considerably high amount of radioactivity (9.95 ± 4.09% ID/g) was concentrated in the kidneys at 45 min p.i. which cleared significantly at 2 h p.i. as 3.78 ± 1.46% ID/g was retained by the kidneys at 2 h p.i. The efficient clearance from the kidneys is possibly associated with the urinary excretion of the radiopeptide. A peptide-based targeting agent that displayed fairly low kidney retention is preferred for diagnostic imaging and particularly for receptor-based radionuclide therapy because of the possible kidney toxicity [5,16]. The low amount of uptake in the bone (up to 0.37 ± 0.14% ID/g) (Table 2) indicates the minimal breakdown of the ^68^Ga-labeled AngII peptide complex in vivo combined with the minor release of free ^68^Ga. These findings are in agreement with the high in vitro metabolic stability observed in the human plasma for the radiolabeled AngII peptide. The accumulation in the extremely vascular organ lungs was also low (up to 1.11 ± 0.37% ID/g) signifying a low trapping by the lungs.

The log *p* value of ^68^Ga-AngII peptide was found to be −0.55 ± 0.13 suggesting its high hydrophilic nature which is apparent from the estimated values of the renal/urinary excretion. A significantly high degree of urinary excretion was noticed for the ^68^Ga-AngII peptide analog, with up to 80.0 ± 12.87% ID found in the urine at 2 h p.i., underlining the importance of hydrophilicity in the renal clearance pattern of a peptide radiopharmaceutical. It is worth mentioning here that we also performed biodistribution of the ^177^Lu-labeled AngII peptide counterpart that displayed more or less the same biological and excretion pattern in Balb/c mice as of ^68^Ga-labeled AngII peptide analog, indicating the insignificant effect of the radionuclides on the pharmacokinetics of the AngII peptide analog under investigation.

### 2.6. In Vivo Tumor Targeting

Given its suitable biodistribution profile, ^68^Ga-labeled AngII peptide analog was further investigated for its in vivo tumor-targeting potential in nude mice carrying TNBC MDA-MB-231 xenografts (Table 2). ^68^Ga-labeled AngII peptide also exhibited efficient and fast clearance from the blood as shown in Balb/c mice, with 0.22 ± 0.10% ID/g radioactivity found in the blood at 2 h p.i. Good and early uptake of 2.18 ± 0.66% ID/g was detected in the MDA-MB-231 tumors, at 45 min p.i., suggesting the potential of the ^68^Ga-labeled AngII peptide for the effective targeting of breast cancer. The tumor showed good retention characteristics and reduced to 1.25 ± 0.37% ID/g at 2 h p.i. (with 43% wash away from the tumors over 2 h). The amount of radioactivity found for the ^68^Ga-labeled AngII peptide analog in the tumors was always more than the radioactivity detected in the blood and muscle. A reasonably good tumor-to-blood and tumor-to-muscle uptake ratios were obtained and are presented in Table 2. A general pattern of better tumor-to-blood and tumor-to-muscle uptake ratios was noticed over time for ^68^Ga-labeled AngII peptide. The tumor-to-blood uptake ratio achieved was 3.96 at 45 min p.i., which improved to 5.68 at 2 h p.i. possibly linked to the fast clearance of the radiolabeled peptide from the blood circulation while the tumor-to-muscle uptake ratio was found to be 5.45 at 45 min, which raised at 2 h p.i. to 6.58.

The amount of radioactivity seen in the major body organs and tissues, such as the lungs, spleen, stomach, liver, and intestines, in nude mice was found to be analogous with normal Balb/c mice (Table 2). Surprisingly, the uptake of radioactivity in the kidneys was significantly lower (4.89 ± 1.97% ID/g) in tumor-bearing nude mice as compared to the one observed in the healthy Balb/c mice (9.95 ± 3.39% ID/g, 45 min p.i.). The exact cause of this low kidney uptake is unclear at present but it may be related to kidneys abnormal function related to heavy tumor burden. The ^68^Ga-labeled AngII peptide analog displayed low bone radioactivity both at 45 min and 2 h p.i. (up to 0.40 ± 0.17% ID/g) suggesting the high in vivo strength of the radiolabeled AngII peptide complex. Alike Balb/c mice, the main pathway of excretion of the ^68^Ga-labeled AngII peptide analog in the tumor-bearing mice was via the renal/urinary system. The amount of the radioactivity excreted into the urine was up to 55% ID, while the clearance of the radiolabeled peptide through the hepatobiliary system (liver + intestines) was insignificant (below 7% ID).

### 2.7. Receptor Specificity Study

To validate the selectivity of the AngII peptide analog for its intended target AT1 receptors, we performed a receptor-blocking assay by administering a blocking dose of ~100 μg of unlabeled AngII peptide premixed with the radiolabeled AngII peptide analog at 2 h p.i. It was found that the blocking dose reduced the uptake in the MDA-MB-231 tumors by approximately 52% (0.60 ± 0.27% ID/g blocked vs. 1.25 ± 0.37% ID/g unblocked, *p* = 0.042), underlining the selectivity of the ^68^Ga-labeled AngII peptide analog for the particular AT1 receptor-expressing breast carcinoma cells. Furthermore, a varying influence pattern of the pharmacological receptor blocking dose was detected mainly in the intestines, heart, and kidneys as the uptake in these organs was reduced to variable degrees depending on the degree of AT1 receptor expression by these organs and tissues [2,17] (Table 2).

### 2.8. In Vivo Metabolic Stability

Athymic nude mice were utilized for the metabolism study to determine the potential radio-metabolites in the urine. An aliquot of urine sample was collected by manual void at the time of sacrifice and analyzed by radio-HPLC. Radio-HPLC evaluation of the urine samples showed one major and one minor HPLC peak (Figure 2). The results signified that most of the radioactivity (>80%) was still bound to the ^68^Ga-AngII peptide analog (Figure 2). There were no major metabolites detected in the urine sample at 2 h p.i. A small hydrophilic metabolite peak was found at about 4 min for the urine sample. This finding confirms that the ^68^Ga-AngII peptide analog is not prone to fast in vivo disintegration and is also in agreement with the results of high in vitro complex strength.

### 2.9. PET Imaging

The preliminary tumor imaging potential of the ^68^Ga-AngII peptide analog was examined in female nude mice carrying subcutaneous triple-negative MDA-MB-231 tumor xenografts at 45 min p.i. While a significant accumulation of radioactivity was seen in the abdominal site together with the urinary bladder, the implanted MDA-MB-231 tumors are reasonably detectable in the PET image (Figure 3), possibly due to the efficient clearance of the radiolabeled peptide from the adjacent organs and tissues. A significantly high urinary bladder radioactivity of ^68^Ga-AngII peptide analog points out its major excretion pathway. The µPET imaging indicates the potential of the ^68^Ga-AngII peptide analog to image the AT1 positive tumors. After the imaging study, quantifiable tissue biodistribution was conducted to validate the observations of the PET imaging. The imaging findings were found to agree with the data acquired in quantifiable biodistribution reported in Table 2.

In sum, the encouraging pharmacokinetics and adequate tumor-targeting profile of the ^68^Ga/^177^Lu-AngII peptide analog revealed the effectiveness of this compound for targeting AT1 receptor-expressing breast carcinoma. The promising findings from this initial study necessitate further exploration of this emerging AngII peptide analog for the diagnostic imaging of breast cancer and possible post-therapy observing of tumor reaction.

## 3. Materials and Methods

### 3.1. General

All analytical grade chemicals and Fmoc-protected amino acids for the peptide synthesis were purchased from different commercial sources and used without further purification. DOTA-tris-(*t*-Bu ester) was purchased from CheMatech, Dijon, France. High-specific activity ^177^Lu radionuclide and ^68^Ge/^68^Ga generator were bought from ITG Isotope Technologies Garching GmbH, München, Germany. The structural identity of the newly synthesized AngII peptide was confirmed by mass spectrometry performed on Agilent 6125 single quadrupole liquid chromatography/mass spectrometry system (LC/MS) (Agilent Technologies, Santa Clara, CA, USA) using an eluent of 0.1% formic acid/29.95% H_2_O/69.95% CH_3_CN at a flow rate of 0.3 mL/min. Reversed-phase high-performance liquid chromatography (RP-HPLC) analyses were performed on a Shimadzu HPLC system (Shimadzu Corporation, Kyoto, Japan) fitted with a dual-wavelength UV-VIS detector (Shimadzu Corporation, Kyoto, Japan), a radioactivity detector (Bioscan, Washington, DC, USA), and the Lauralite chromatogram analysis software (LabLogic Systems Ltd., Version 4.2.4., Sheffield, UK). Radioactive samples were measured by using a γ-counter (Mucha, Raytest Isotopenmessgeräte GmbH, Straubenhardt, Germany).

### 3.2. Synthesis of the AngII Peptide

Synthesis of AngII peptide was performed with Fmoc–rink amide-linker functionalized polystyrene resin (100–200 mesh) on a 0.1 mmol scale by typical procedures of solid-phase peptide synthesis using peptide synthesis glass reaction vessel (Peptides International, Louisville, KY, USA) as previously described [18,19]. In brief, starting at the C-terminus, the first amino acid, (4-times molar excess as compared to the resin) was preactivated at its α-carboxylic function using an activating reagent, such as HATU (Hexafluorophosphate Azabenzotriazole Tetramethyl Uronium) in the presence of a base DIEA (diisopropylethylamine) and allowed to react for 60 min at room temperature with the resin amino group. After completion of the coupling reaction as confirmed by the Kaiser ninhydrin test, the Fmoc-protecting group was cleaved by treatment with 20% piperidine solution in DMF (*N*,*N*-dimethylformamide). The peptide sequence was then elongated in repeated cycles of Fmoc deprotection (with 20% piperidine in DMF), activation (reaction of 4-fold molar excess of subsequent Fmoc-protected amino acids with HATU and DIEA), and coupling of the preactivated amino acids with the peptide-resin. Once all the desired amino acids were added to the peptide resin, the Fmoc-group at the N-terminal was removed to facilitate its attachment with a DOTA chelating group.

DOTA-tris-(*t*-Bu ester) was coupled to the NH_2_ terminus of the peptide on the resin, with a 2.5-fold excess to the resin, was first preactivated with HATU (2.5 equivalent) and DIEA (5 equivalent) for 15 min before the manual conjugation with the AngII peptide sequence. The coupling reaction was carried out overnight and its completion was assessed by the negative Kaiser test. Finally, the AngII peptide was cleaved from the resin together with other side-chain protecting groups by treating it with a cleavage cocktail (~4 mL) of 95% trifluoroacetic acid (TFA), 2.5% water and 2.5% triisopropylsilane, TIS) for 4 h at room temperature. The resin was removed by filtration and TFA was evaporated by rotavap. The DOTA-AngII peptide was then collected by precipitation with cold diethyl ether and its purity and identity were determined by HPLC and mass spectrometry.

### 3.3. Radiolabeling of Angiotensin II Peptide with ^68^Ga and ^177^Lu

*Radiolabeling with ^68^Ga*: The labeling method is basically as described previously [18,19]. Briefly, ~25 µg of peptide conjugate (1 mg/mL H_2_O/CH_3_CN (1:1 *v*/*v*) solution) was mixed with 150 µL of 2.5 M sodium acetate buffer. To this 100 µL of EtOH was added. This was followed by the addition of [^68^Ga]GaCl_3_ in 0.05 M HCl (111–185 MBq; 3–5 mCi, 1 mL) (eluted from ^68^Ge–^68^Ga generator, ITG Germany), and adjusting the pH ~4.5. The labeling mixture was then heated for 15 min at 90 °C to enhance labeling kinetics. The reaction mixture was cooled to room temperature and filtered through a 0.2-μm pore syringe filter before HPLC analysis.

*Radiolabeling with ^177^Lu*: The labeling method is mostly as described previously [18,19]. Briefly, ~50 µg of peptide conjugate (1 mg/mL H_2_O/CH_3_CN (1:1 *v*/*v*) was mixed with ~200 µL of 0.5 M ammonium acetate buffer. To this, 40 µL gentisic acid (40 mg/mL aqueous solution) followed by ~100 µL of [^177^Lu]LuCl_3_ (74–148 MBq; 2–4 mCi), dissolved in 0.04 M HCl, was added. The reaction mixture with a final pH of 4.5 was heated at 90 °C for 30-min to accelerate labeling kinetics. After cooling, the preparation was filtered through a 0.2-μm pore syringe filter before HPLC analysis.

A nonradioactive gallium complex, ^nat^Ga-DOTA-AngII peptide was prepared by reacting DOTA-AngII peptide analog (0.5 μM, 1 equiv.) with Ga(III)Cl_3_ (1.0 μM, 2 equiv.) dissolved in 400 μL of 0.05 M HCl. To this 400 μL of 0.2 M ammonium acetate buffer (pH 5.0) was added. The reaction mixture was heated for 60 min at 90 °C to enhance the complex formation.

### 3.4. HPLC Analysis

Reversed-phase HPLC analysis was carried out on a Shimadzu HPLC system (Shimadzu, Japan) as described previously [18,19]. In summary, the HPLC analysis and purification were performed on a Shimadzu HPLC system using a Grace Alltima C18 reversed-phase column (10 μm, 250 × 4.6 mm). For all HPLC experiments, a gradient system of 0.1% (*v*/*v*) TFA in water (solvent A) and 0.1% (*v*/*v*) TFA in CH_3_CN (solvent B) at a flow rate of 1.1 mL/min was used. The HPLC gradient starts with a solvent composition of 95% A and 5% B from 0 to 2.5 min followed by a linear gradient of 95% A and 5% B to 5% A and 95% B over 30 min. The gradient remained at this position for 3 min before switching back to initial settings of 95% A and 5% B for another 7 min. The major peak of the AngII peptide was collected and the organic solvent was then slowly evaporated under a stream of nitrogen gas. Radiochemical purity was estimated by evaluating the radioactivity peak eluted and calculating the area under the peak (ROI). The HPLC-purified AngII peptide was formulated in sterile saline and utilized for in vitro and in vivo assays.

### 3.5. Measurement of Partition Coefficient

For lipophilicity determination, the HPLC-purified radiolabeled peptide (~25 μL, 0.925 MBq; 25 μCi) was added into a glass tube containing an equal volume mixture of n-octanol and water (1 mL each). The samples were vortexed for 5 min and subsequently centrifuged (5000 rpm, 5 min) to yield two immiscible layers. Duplicate samples (100 μL) from each layer were carefully taken (to avoid cross-contamination between the layers) for radioactivity measurement using a γ-counter. The partition coefficient was determined by the function: partition coefficient = Log10 (radioactivity in octanol layer/radioactivity in aqueous layer) [18].

### 3.6. Stability in Human Plasma

Human plasma was obtained from the institutional blood bank from volunteers’ blood donors (with consent). The metabolic stability of ^177^Lu-DOTA-AngII peptide was determined following a method essentially as described previously [18,19]. In summary, the radiolabeled peptide (50 μL) was mixed with plasma (400 μL) and incubated in duplicate at 37 °C for up to 5 h. After incubation, the plasma proteins were precipitated using a mixture of CH_3_CN/EtOH (1:1 *v*/*v*, 400 μL). The supernatant layer was collected by centrifugation (5000 rpm for 5 min), filtered through a 0.2-µm Millex GP filter, and analyzed by radio-HPLC to assess the product stability.

### 3.7. In Vitro Cell Binding and Internalization

Methods for preparing the cells were as described previously [19]. In brief, breast cancer cell lines MDA-MB-231 and MCF7 (American Type Culture Collection, Rockville, MD, USA) were grown as monolayers at 37 °C in a humidified atmosphere containing RPMI-1640 culture media with 10% fetal bovine serum (FBS) in the tissue culture flasks. Twenty-four hours before conducting the tumor implantation, the media was replaced with RPMI-1640/10% FBS. The cells were grown to 80–90% confluency and harvested by trypsinization. After centrifugation, ~50 million cells were suspended in 2 mL PBS. For inoculation per nude mouse, ~5 million MDA-MB-231 cells in 200 μL PBS were used.

The binding of radiolabeled AngII peptide to TNBC MDA-MB-231 and estrogen receptor-positive MCF7 human breast cancer cell lines (obtained from the ATCC, Rockville, MD, USA) and subsequent internalization into these cancer cell lines were carried out as described previously [18,19]. In brief, various amounts of the AngII peptide ranging from 1.0 to 100 nM (prepared from the serial dilutions of HPLC-purified peptide) in the presence of ^68^Ga/^177^Lu-AngII peptide (~111 kBq, 3 µCi) were mixed with the fixed amount of ~200,000 cells (in 200 μL phosphate-buffered saline, PBS) and incubated in duplicate at 4 °C for 90-min with occasional shaking. The initial concentration of radiolabeled AngII peptide was determined following a known HPLC technique with simultaneous detection by UV absorbance [20,21]. Since the amount of the AngII peptide in the reaction mixture was too small to be detected by the UV absorption at 220 nm, the radioactive peptide chromatograms were compared (with UV chromatograms) with those of known concentrations of the corresponding nonradioactive AngII peptide for the mass estimation [20,21]. Incubation was terminated by the dilution with cold PBS (200 μL) followed by centrifugation. The supernatant was collected and cell pellets were rapidly washed with cold PBS to remove any unbound peptide. Radioactivity in the cell pellet (total bound) and the supernatant (unbound) was measured in a γ-counter. Nonspecific binding was determined in the presence of approximately 200-fold excess of unlabeled AngII peptide. Specific binding is calculated by subtracting the non-specifically bound radioactivity from that of the total binding. The dissociation constant (*K*_d_) is calculated using a plot of cell-bound activity versus the concentrations of the AngII peptide using the GraphPad Prism software version 5.03 (GraphPad Software Inc., San Diego, CA, USA).

Thereafter, the cell pellet was incubated with 300 μL of the acidic buffer (0.02 M sodium acetate in saline, pH 5.0) [15,19] for 15 min at 37 °C to allow internalization of the surface-bound radiotracer into cancer cells. Following incubation, the cells were separated by centrifugation and washed with a cold acidic buffer. The amount of cell surface-bound (acid-wash) and intracellular radioactivity (acid-resistant) was determined by measuring the radioactivity of the supernatant and the cell pellet, respectively, in a γ-counter.

### 3.8. In Vivo Animal Biodistribution and Tumor Targeting

Approval from the Institutional Animal Care and Use Committee was obtained for the animal protocol used. Additionally, strict international regulations govern the safe and proper use of laboratory animals employed during animal experiments [22]. In vivo, biodistribution was performed on healthy female mice according to the methods essentially described previously [18,19]. In summary, Balb/c mice (*n* = 5 per time point, body mass 20–25 g) at 45 min and 2 h were administered with the HPLC-purified AngII peptide (~1 µg, 100 µL, 0.925–1.85 MBq; 25–50 µCi) via lateral tail vein injection. At the specified time points, the animals were sacrificed by cervical dislocation. A fraction of the blood was collected from cardiac puncture. Urine was also collected and measured with the bladder contents. Major organs, such as the lungs, pancreas, stomach, liver, heart, kidneys, and intestines were isolated, weighed, and measured for radioactivity. The uptake values of the radiolabeled peptide in different organs are expressed as the percent injected dose per gram (% ID/g) of tissue/organ. The radioactivity in the urine with bladder contents is shown as the percent of the injected dose per organ (% ID/organ). To calculate the injected dose, a standard (20-fold dilution of injected dose) was prepared and counted together with animal tissue samples.

In vivo, tumor uptake was conducted as reported previously [18,19]. In summary, ~5 million MDA-MB-231 cells in 200 µL sterile PBS were implanted subcutaneously into each nude mouse. Once tumor growth became sufficient (~50–500 mg), tumor uptake and organ biodistribution were carried out after injecting the radiolabeled AngII peptide. The percent of the injected dose per gram in the tumors was determined by using a custom-designed Quattro Pro X9 spreadsheet (Corel Corporation, Austin, TX, USA).

For the receptor-blocking study, an additional group of mice was injected with the radiolabeled AngII peptide premixed with unlabeled AngII peptide (~100 μg) to serve as a receptor-blocking agent. In vivo, biodistribution and blocking studies were performed at 2 h p.i. as stated before.

To determine the radio-metabolites of the ^68^Ga-labeled AngII peptide analog, an aliquot of urine sample was collected by manual void at the time of sacrifice and mixed with an equal volume of CH_3_CN. The mixture was centrifuged at 7000 rpm for 5 min. The resulting supernatant was collected and passed through a 0.20 μm Millex-LG filter unit to remove any precipitate or foreign particles and then analyzed by radio-HPLC.

### 3.9. Micro-PET Imaging

Micro PET imaging was done as described before with minor modifications [19]. To find out the tumor-targeting potential, ^68^Ga-labeled AngII peptide analog (~1.85–2.78 MBq; 50–75 µCi,100 µL) was administered through the tail vein injection into a nude mouse carrying MDA-MB-231 tumor xenografts. A total of 15 min of static images, with 10 frames (1.5 min each), were obtained in the prone position using a mini PET camera (Bioemtech, Athens, Greece). The PET camera is equipped with a pixelated BGO (bismuth germanate) scintillator and 4 arrays of compact position-sensitive photomultiplier tubes, with a field of view of 48 mm × 98 mm and an intrinsic spatial resolution of ~1.5 mm. The manufacturer preinstalled concurrent image reconstruction software Visual Eyes version 2.01 was used to convert the acquired frames into 2D image visualization. Images were then generated using Image J software version 1.46r (National Institute of Health, Bethesda, MD, USA). Following imaging, animals were dissected and quantitative biodistribution was performed to validate the findings of the PET imaging.

### 3.10. Statistical Analysis

Experimental data are represented as mean ± S.D. where appropriate. For data evaluation, mean values were compared using the Student’s *t*-test (GraphPad Software version 5.03, San Diego, CA, USA), and a probability value (*p*) ˂ 0.05 was considered statistically significant.

## 4. Conclusions

The present piece of work describes for the first time the preparation of the AngII peptide analog by the solid-phase synthesis and investigation of a novel ^68^Ga/^177^Lu-labeled AngII peptide analog in preclinical settings. The AngII tumor-targeting peptide displayed high metabolic stability. The cell binding for respective breast cancer cell lines was saturable and receptor-specific. The radiolabeled AngII peptide exhibited rapid pharmacokinetics and good tumor-targeting potential in MDA-MB-231 tumor xenografts models. Still more preclinical evaluations are needed to determine the real effectiveness of this new and emerging class of angiotensin peptides for tumor detection; however, the outcome of this study can be beneficial for the designing of more efficacious peptide-based agents for the efficient targeting of AT1 receptor-positive tumors.

## Figures and Tables

**Figure 1 pharmaceuticals-16-01550-f001:**
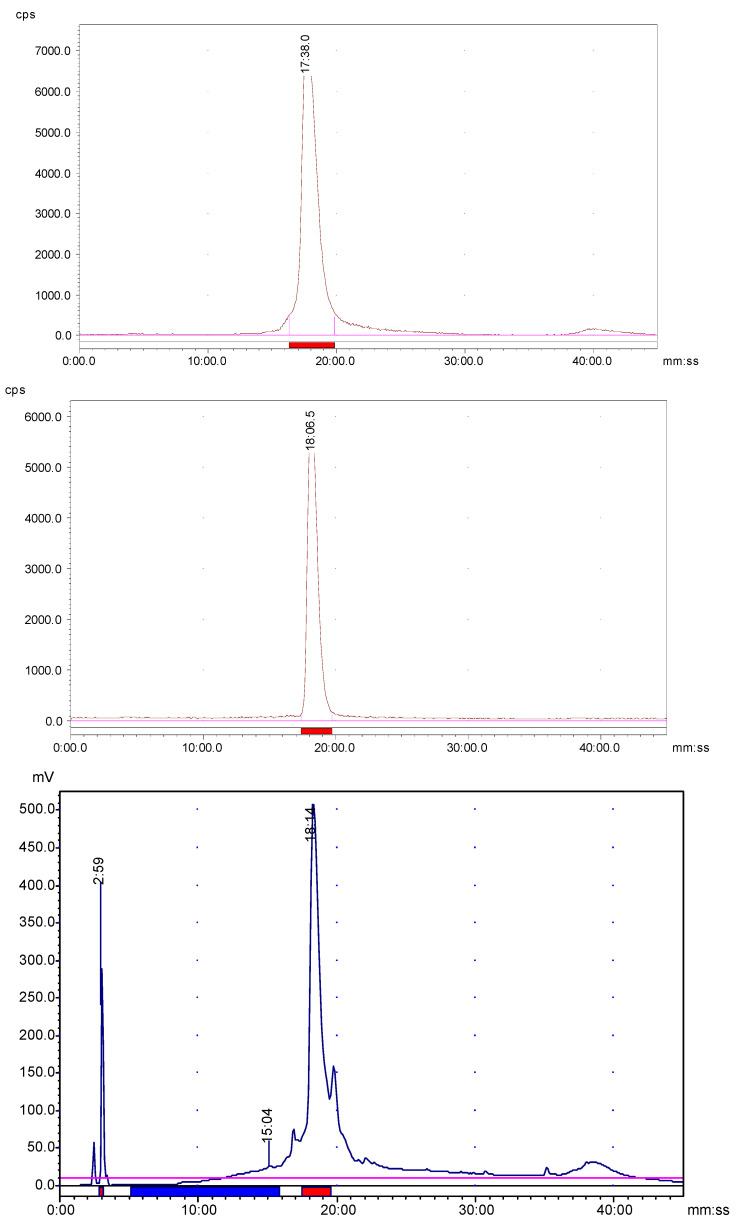
Representative HPLC elution profiles of the ^68^Ga-DOTA-AngII peptide (**upper**); ^177^Lu-DOTA-AngII peptide (**middle**); and the reference DOTA-AngII peptide (**lower**) labeled with cold ^nat^Ga for identity confirmation (UV: 220 nm) after 60 min post-labeling. The radiochemical purity was always greater than 85% after post-labeling chromatographic purification. Note: The UV detector is connected before the radioactive detector.

**Figure 2 pharmaceuticals-16-01550-f002:**
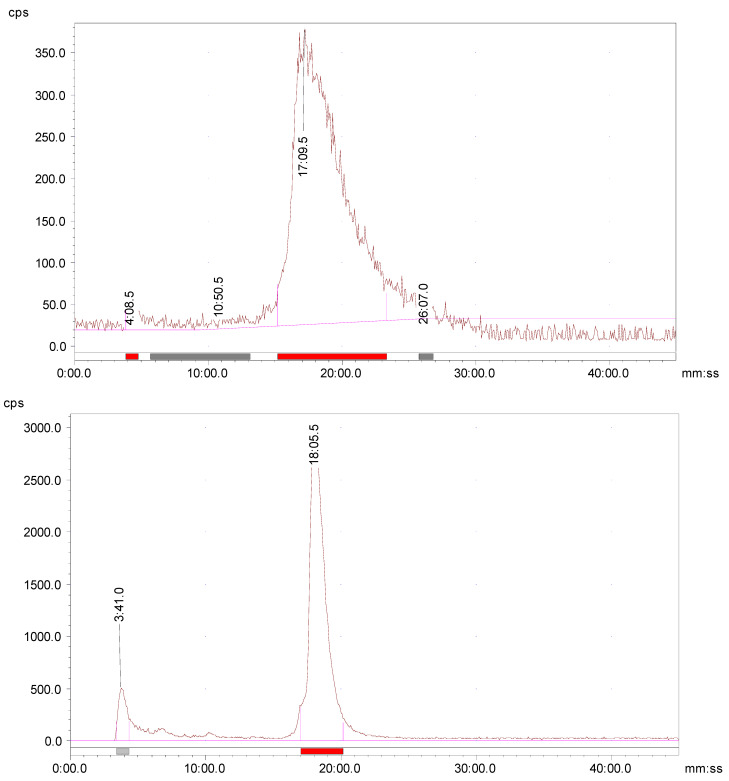
Radio-HPLC analysis of a mouse urine sample collected at 2 h p.i. after injecting the ^68^Ga-DOTA-AngII peptide analog (**upper**); Radio-HPLC chromatogram of ^177^Lu-DOTA-AngII peptide analog after 5 h incubation with human plasma (**lower**).

**Figure 3 pharmaceuticals-16-01550-f003:**
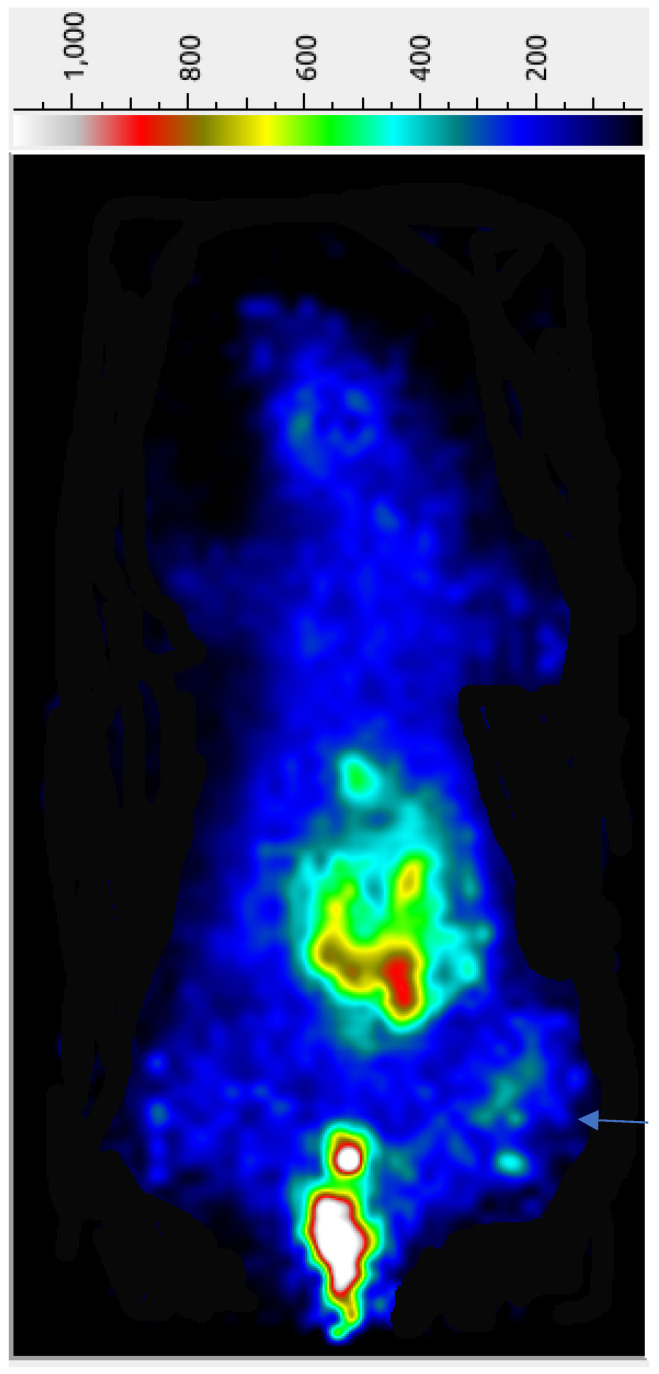
Micro-PETcamera image of a female nude mouse, with MDA-MB-231 breast cancer xenografts, after the tail vein injection of ~75 µCi of ^68^Ga-DOTA-AngII peptide analog at 45 min post-injection. The arrow indicates the tumor position.

**Table 1 pharmaceuticals-16-01550-t001:** Evaluation of cell-binding affinity and cellular internalization of ^68^Ga/^177^Lu-labeled AngII peptides to MDA-MB-231 and MCF-7 human breast cancer cell lines. Cell binding affinity was determined after 90 min incubation of radiolabeled peptide with the breast cancer cells. Experiments were repeated three times; Mean values ± SD.

	Cell Line	*K*_d_ (nM)	% Internalization
^68^Ga-AngII	MDA-MB-231	30.0 ± 6.88	−
	MCF-7	39.25 ± 8.10	−
^177^Lu-AngII	MDA-MB-231	27.73 ± 5.97	14.97 ± 5.03
	MCF-7	35.14 ± 7.09	11.75 ± 3.70

**Table 2 pharmaceuticals-16-01550-t002:** In vivo tissue biodistribution of ^68^Ga-DOTA-AngII peptide in normal Balb/c mice and tumor targeting in nude mice bearing MDA-MB-231 tumor xenografts. Experiments were conducted at 45 min and 2 h post-injection (*n* = 4–6). Data are shown as % injected dose per gram of tissue (mean values ± SD). Urinary excreted values are given as % injected dose per tissue. A blocking assay was performed at 2 h post-injection by co-injecting ~100 µg of unlabeled AngII peptide analog. * A portion of the intestines without their contents was measured.

	Balb/c Mice		Nude Mice		
	45 min	2 h	45 min	2 h	Blocked
Blood	0.85 ± 0.24	0.61 ± 0.15	0.55 ± 0.12	0.22 ± 0.10	0.85 ± 0.20
Lungs	1.11 ± 0.37	0.60 ± 0.20	0.57 ± 0.16	0.39 ± 0.15	0.89 ± 0.35
Spleen	0.51 ± 0.10	0.31 ± 0.09	0.39 ± 0.08	0.24 ± 0.05	0.35 ± 0.09
Stomach	1.06 ± 0.41	0.81 ± 0.23	0.34 ± 0.10	0.30 ± 0.08	0.49 ± 0.18
Pancreas	0.50 ± 0.11	0.33 ± 0.06	0.47 ± 0.11	0.29 ± 0.05	0.54 ± 0.12
Liver	1.54 ± 0.28	0.48 ± 0.17	0.77 ± 0.22	0.65 ± 0.28	0.85 ± 0.24
Intestines *	1.45 ± 0.39	1.0 ± 0.28	1.20 ± 0.35	1.05 ± 0.31	0.80 ± 0.20
Muscle/bone	0.37 ± 0.14	0.28 ± 0.10	0.40 ± 0.17	0.19 ± 0.10	0.24 ± 0.10
Kidneys	9.95 ± 3.39	3.78 ± 1.46	4.89 ± 1.97	3.97 ± 1.20	2.01 ± 0.87
Heart	1.39 ± 0.51	0.99 ± 0.26	0.88 ± 0.30	0.65 ± 0.19	0.23 ± 0.10
Urine + Bladder (% ID)	67.0 ± 8.90	80.0 ± 12.87	49.90 ± 10.87	55.0 ± 8.79	43.45 ± 7.90
Tumor	−	−	2.18 ± 0.66	1.25 ± 0.37	0.60 ± 0.27
*Uptake ratios*					
Tumor/blood	−	−	3.96	5.68	
Tumor/muscle			5.45	6.58	

## Data Availability

Data is contained within the article.

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
