# Peer review of "Preparation, Radiolabeling with 68Ga/177Lu and Preclinical Evaluation of Novel Angiotensin Peptide Analog: A New Class of Peptides for Breast Cancer Targeting"

_pharmaceuticals, 2023, doi:10.3390/ph16111550_

Round 1

Reviewer 1 Report

Results & discussion: line99-122 is repeating introduction part and should be combined in introduction

Line 139: 68GaCl3 should be [68Ga]GaCl3, please check all other nomenclature and make sure it is written correctly.

Line 144: Please use the correct term, it should be molar activity instead of “specific radioactivity”

Line 145: Please always use Becquerel (international unit) instead of Ci (you can mention Ci in bracket if you want), check the whole paper

Figure 1,2: Can you please put the radiochemical yield on the HPLC profiles?

Figure 2: Can you please integrate the area more accurately? 

Author Response

  1. Line 99-122: now combined with the Introduction
  2. 68GaCl3 is now changed to [68Ga]GaCl3
  3. Specific radioactivity is changed to molar radioactivity
  4. Bq is mentioned instead of Ci where appropriate
  5. Radiochemical yield is mentioned in Figure 1
  6. the integrated area is quite accurate, hard to change now

Reviewer 2 Report

The manuscript submitted by Subhani M. Okarvi discusses the radiolabeling and evaluation of a novel angiotensin peptide for breast cancer imaging. The author details the peptide synthesis, which yielded favorable results, the radiolabeling of the peptide with Ga-68 and Lu-177, and the characterization of the radiolabeled peptide's binding potency. Additionally, both in vitro and in vivo studies demonstrated the effectiveness of the Ga-68 radiolabeled peptide. The author has exhibited commendable dedication to this research, resulting in a well-structured manuscript. Nevertheless, several key points warrant attention before acceptance.

Comments:

  1. In the Introduction section, it would be beneficial for the author to include background information on PET imaging of AT1.
  2. The content spanning from Line 99 to 116 appears overly detailed and could benefit from greater conciseness.
  3. In Line 144, please replace "specific radioactivity" with "molar radioactivity" for accuracy.
  4. In Figure 1, consider integrating the peak around ~40 minutes to enhance clarity.
  5. Figure 3 requires improved analysis of the PET image results; kindly include an arrow to pinpoint the tumor's location, co-localize with CT image will be better.
  6. In Part 3.6, the experimental protocol lacks precision regarding the radioactivity addition. Please specify this crucial detail.
  7. The manuscript's title should be revised to accurately reflect its content.

Author Response

  1. All available information about PET imaging of AT1 is included in the Introduction section.
  2. The contents of lines 99-116 are now combined with the Introduction
  3. Line 144: specific radioactivity is now replaced with molar radioactivity
  4. It is hard to integrate the peak at around 40 minutes.
  5. An arrow is added to indicate the location of tumors. Our mini PET camera is somewhat low-resolution and without the CT co-localization option.
  6. In 3.6., the radioactivity addition is included.

Reviewer 3 Report

The current study introduces a novel AngII peptide analog synthesized through solid-phase synthesis and evaluates it with 68Ga/177Lu labeling in preclinical settings. This peptide exhibits excellent metabolic stability and selective binding to breast cancer cell lines. In in vivo studies, it demonstrates rapid pharmacokinetics and exhibits tumor targeting in MDA-MB-231 xenograft models. Overall, this study suggests the potential of 68Ga/177Lu-DOTA-AngII peptide for theranostic applications in breast carcinomas, emphasizing its promise as a novel peptide class for tumor targeting. However, several additional studies may need consideration before publication:

1.  Tumor uptake appears significanty low and less visible in PET imaging. This contrasts with the high nanomolar affinity observed in vitro. The authors should investigate the reasons behind this discrepancy, considering the claimed stability of the molecule in vivo and the reasonable specific activity. Could the difference be attributed to the 90-minute incubation in cell studies and the 45-minute imaging time? BioD data suggests a different story, so further investigation is warranted. Perhaps MDA-MB-231 is not an ideal model to demonstrate the application's effectiveness. Conducting a cell uptake study with different cell lines, including negative and blocking controls, may shed light on the binding differences.

2.  The study reports a log P value of -0.55 ± 0.11 for the 68Ga-labeled AngII peptide analog, indicating its hydrophilic nature. It would be beneficial to include background research supporting this characterization.

3.  The synthesis of 68Ga-DOTA-AngII peptide is discussed, but details about the synthesis of a cold standard and the retention time of DOTA-Ga under the same HPLC conditions are missing.

In addition to these points, minor comments include the suggestion to extract raw HPLC data in excel format for better representation, clarification regarding the number of animals used in PET imaging, and the inclusion of methods for determining specific activity and log P in the manuscript.

Author Response

  1. The poor visibility of the tumor uptake in PET imaging is possibly related to the somewhat poor resolution of the mini PET camera which provides images in 2D format. In addition to the MDA-MB-231 cell line, the MCF7 cell line was also tested in this study
  2. The log P characterization method is now included in the text.
  3. The cold standard of DOTA-AngII with the retention time is included in the text and also in Figure 1.

Round 2

Reviewer 3 Report

The manuscript can be accepted for publication.